# JOINTLY LEARNING VISUAL AND AUDITORY SPEECH REPRESENTATIONS FROM RAW DATA

**Alexandros Haliassos**[1][*] **Pingchuan Ma**[1] **Rodrigo Mira**[1] **Stavros Petridis**[1,2] **Maja Pantic**[1,2]

[1]Imperial College London       [2]Meta AI

`alexandros.haliassos14@imperial.ac.uk`

## ABSTRACT

We present RAVEn, a self-supervised multi-modal approach to jointly learn visual and auditory speech representations. Our pre-training objective involves encoding masked inputs, and then predicting contextualised targets generated by slowly-evolving momentum encoders. Driven by the inherent differences between video and audio, our design is *asymmetric* w.r.t. the two modalities' pretext tasks: Whereas the auditory stream predicts both the visual and auditory targets, the visual one predicts only the auditory targets. We observe strong results in low- and high-resource labelled data settings when fine-tuning the visual and auditory encoders resulting from a *single* pre-training stage, in which the encoders are jointly trained. Notably, RAVEn surpasses all self-supervised methods on visual speech recognition (VSR) on LRS3, and combining RAVEn with self-training *using only 30 hours* of labelled data even outperforms a recent semi-supervised method trained on *90,000* hours of non-public data. At the same time, we achieve state-of-the-art results in the LRS3 low-resource setting for auditory speech recognition (as well as for VSR). Our findings point to the viability of learning powerful speech representations entirely from raw video and audio, *i.e.*, without relying on handcrafted features. Code and models are available at `https://github.com/ahaliassos/raven`.

## 1 INTRODUCTION

The sound of someone articulating words coincides with the sight of movements in and around their mouth. Both a recording of a speech waveform and a corresponding silent video of mouth motion provide rich - but not identical - information on which words were uttered. Despite the difficulty of interpreting lip movements compared with an audio waveform, the task of visual speech recognition (VSR; also known as lipreading) has important applications, ranging from recognising utterances in a noisy environment (Ma et al., 2021b; Afouras et al., 2018a; Martinez et al., 2020; Makino et al., 2019) and aiding people suffering from aphonia (an inability to speak), to transcribing archival silent films and detecting DeepFake videos (Haliassos et al., 2021).

Auditory (also known as automatic) speech recognition (ASR) and VSR benefit greatly from the combination of high-capacity neural networks and large datasets. Rapid advances of modern hardware are enabling the use of ever-growing, data-hungry networks, but the effort required for transcription hinders the scaling of labelled data along with the models. One way to leverage unlabelled videos for VSR is to use an external ASR model for pseudo-labelling (Afouras et al., 2020; Ma et al., 2022). However, this requires a large amount of labelled data to train a strong ASR model in the first place, and supervised VSR training with long sequences often poses optimisation problems, requiring costly curriculum learning strategies (Chung et al., 2017; Ma et al., 2022) or pre-training the feature extractor with isolated words (Afouras et al., 2018a; Ma et al., 2021b).

A solution is to first learn, in a self-supervised way, general representations from large corpora of unlabelled data, and then fine-tune them on smaller labelled datasets (Mohamed et al., 2022). The fine-grained correspondence between the (synchronised) visual and auditory modalities provides a

---

[*]Work done at Meta AI.

natural source of self-supervision, and can produce highly semantic representations invariant to noise not shared between the modalities. However, approaches leveraging this correspondence either (1) only work for word-level samples rather than continuous speech (Chung & Zisserman, 2016; Chung et al., 2019; 2020); (2) use handcrafted features (*e.g.,* spectrograms or MFCCs) as their inputs or targets (Ma et al., 2021a; Shi et al., 2022), which contain inductive biases that may influence the learned representations; (3) use multi-stage pre-training procedures (Ma et al., 2021a; Shi et al., 2022; Pan et al., 2022); and/or (4) use separate pre-training strategies for VSR and ASR (Shi et al., 2022), complicating the process of obtaining representations suitable for both tasks.

In this paper, we present a single-stage self-supervised approach that jointly learns visual and auditory speech representations from raw video and audio only. We dub our approach RAVEn (**R**aw **A**udio-**V**isual Speech **En**coders). It involves a pair of student-teacher networks for each modality, whereby the students encode temporally-masked inputs, and, through the use of lightweight Transformer-based predictors, regress outputs of momentum-based teachers (Grill et al., 2020; Caron et al., 2021) that are presented with unmasked inputs. Further, given that audio contains more information relevant to speech than video, we propose a learning strategy that accounts for the expected difference in the quality of targets between the modalities. Namely, while the audio student predicts outputs from both video and audio teachers (cross- and within-modal learning), the video student predicts only auditory targets (cross-modal learning). As we show, this setup leads to better downstream performance for both VSR and ASR as opposed to other strategies.

We conduct experiments with models and datasets of different sizes. We find that, when fine-tuning our pre-trained models for VSR and ASR with only 30 hours of labelled data from LRS3 (Afouras et al., 2018b), RAVEn surpasses recent self-supervised methods by a large margin in most settings. Coupling pre-training with self-training reaches 23.8% WER for VSR on LRS3, even outperforming a method trained on *3000×* more transcribed hours (Serdyuk et al., 2021). At the same time, we are better than or on par with the recent AV-HuBERT method (Shi et al., 2022) on ASR, without using a task-dependent pre-training strategy nor handcrafted features. Using the full 433-hour LRS3 dataset for fine-tuning pushes the results even further, achieving 23.1% / 1.4% WER for VSR / ASR, respectively. Similarly strong performance is observed on the LRS2 dataset (Appendix B).

## 2 RELATED WORK

**Masked prediction.** The pre-training task of predicting missing content given masked inputs has proven successful in various domains, such as natural language processing (Devlin et al., 2019; Radford et al., 2018; 2019; Brown et al., 2020), image recognition (He et al., 2021; Xie et al., 2022; Bao et al., 2021), and speech recognition (Baevski et al., 2020; Hsu et al., 2021; Shi et al., 2022). An important aspect of masked prediction is the nature of the targets. Some works (He et al., 2021; Xie et al., 2022) use pixels as targets; others use pre-trained tokenisers (Bao et al., 2021) or modality-specific handcrafted features (Wei et al., 2021; Hsu et al., 2021; Shi et al., 2022). Our method, in contrast, generates targets from *raw video and audio* using momentum encoders.

**Self-distillation for unsupervised representation learning.** RAVEn is partly inspired by the success of self-distillation in self-supervised learning with visual data (Grill et al., 2020; Caron et al., 2021); such works target invariance w.r.t. image-specific augmentations. In contrast, RAVEn does not rely on domain-specific augmentations but rather on a combination of cross-modal learning and masked prediction to drive representation learning for visual and auditory speech signals. data2vec (Baevski et al., 2022) combines masked prediction with a momentum encoder, but, aside from being uni-modal, it is different methodologically in multiple ways. For example, it applies ad-hoc normalisation and averaging techniques to the targets to prevent representation collapse, while our targets are simply the outputs of the encoders, which are regressed via Transformer-based heads.

**Self-supervised audiovisual learning.** Audiovisual correspondence has been used to learn global representations for action recognition through the use of clustering (Alwassel et al., 2020; Asano et al., 2020), contrastive learning (Arandjelovic & Zisserman, 2017; 2018; Korbar et al., 2018; Patrick et al., 2020; Morgado et al., 2021; Ma et al., 2021c), or representation matching (Recasens et al., 2021). Cross-modal learning has also found uses in biometric matching (Nagrani et al., 2018a;b), emotion recognition (Shukla et al., 2021), and DeepFake detection (Haliassos et al., 2022). We employ cross- and within-modal losses to learn temporally-varying speech representations.

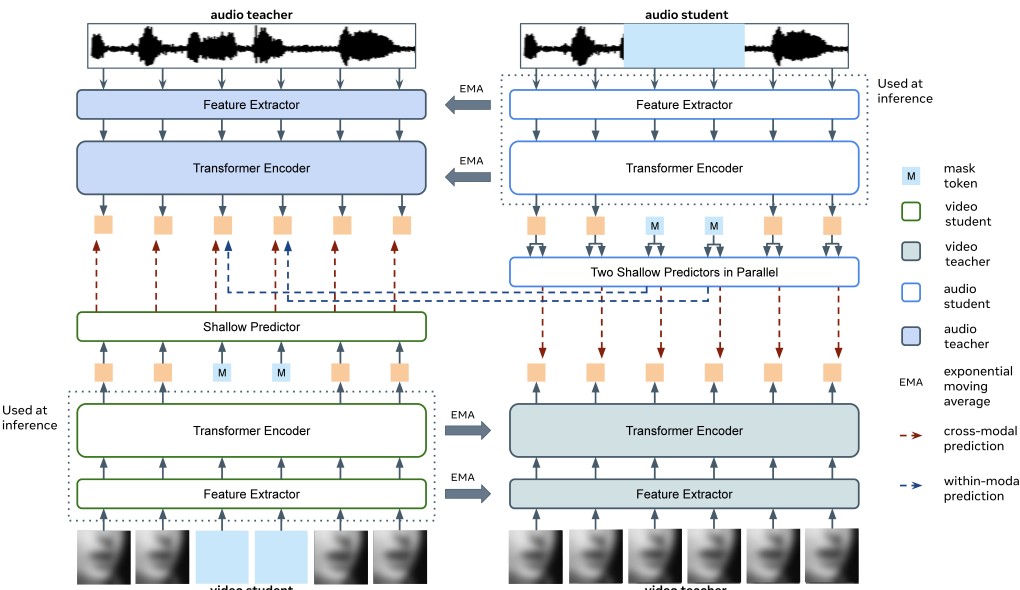

Figure 1: **RAVEn overview**. Given masked video and audio, students predict outputs of unmasked momentum teachers, via shallow Transformer predictors that intake mask tokens. The audio student predicts outputs from both audio and video teachers; the video student predicts only audio targets. Cross-modal losses are applied on all features; the within-modal loss is computed only on masked features. Only the student encoders are fine-tuned for VSR/ASR. Frames blurred for anonymity.

**Self-supervised audiovisual learning for speech recognition.** Earlier audiovisual self-supervised works (Chung & Zisserman, 2016; Chung et al., 2019; 2020) tended to focus on word-level lipreading. Recently, some attention has been paid to the more realistic task of *continuous* speech recognition (Ma et al., 2021a; Shi et al., 2022; Sheng et al., 2021; Pan et al., 2022; Ma et al., 2021c). Sheng et al. (2021); Ma et al. (2021c) use contrastive learning and apply their method to VSR. Ma et al. (2021a) predict speech features using an external PASE+ encoder (Ravanelli et al., 2020). Pan et al. (2022) transfer pre-trained visual and auditory encoders, which were separately trained via contrastive losses. AV-HuBERT (Shi et al., 2022) predicts iteratively-refined cluster assignments from masked audiovisual inputs and achieves impressive VSR and ASR performance. It employs multiple stages of alternating between offline clustering and cluster assignment prediction, and relies on hand-crafted audio features (MFCCs) for cluster initialisation, which is shown to be crucial to the performance (Shi et al., 2022). We demonstrate that it is possible to jointly learn effective representations for VSR and ASR in a single stage simply from raw video and audio.

## 3 METHOD

### 3.1 PRE-TRAINING

Our architecture consists of a pair of student-teacher networks per modality (see Figure 1). The students intake masked inputs and predict targets formed by teachers receiving unmasked inputs.

**Masking.** We employ masking to encourage the students to take context into account when solving the task. Given a grayscale video $x^v \in \mathbb{R}^{T \times H \times W}$ with resolution $(H, W)$ and $T$ frames, and an audio sample $x^a \in \mathbb{R}^N$ of length $N$, we randomly sample with probability 0.2 each video frame to be the starting mask index, and if selected, then the consecutive three frames are zeroed out (ablation in Section 4.4). A similar mask is applied to the auditory input, except that it is enlarged by a factor of $16\,000/25 = 640$ (since the audio is sampled at 16,000 Hz and the video at 25 fps).

**Encoders.** The masked video and audio, $\bar{x}^v$ and $\bar{x}^a$ respectively, are fed to their corresponding student encoders $f_e^v$ and $f_e^a$, yielding features $f_e^v(\bar{x}^v) \in \mathbb{R}^{T \times D}$ and $f_e^a(\bar{x}^a) \in \mathbb{R}^{T \times D}$, where $D$ is

the dimensionality of each feature. Both $f_e^v$ and $f_e^a$ consist of a modality-specific, convolutional feature extractor followed by a temporal encoder, as in related VSR/ASR works (Ma et al., 2022; Baevski et al., 2020). The video feature extractor is a 2D ResNet18 (He et al., 2016) with a 3D convolutional stem (Petridis et al., 2018), outputting an embedding per frame. On the audio side, we use a 1D ResNet18 which produces features at 25 fps, to match the video sampling rate (see Appendix A.5). The temporal encoder for each modality is a Transformer (Vaswani et al., 2017) (without a classification head) with hidden size $D$. We use relative positional embeddings (Dai et al., 2019), and following Chen et al. (2021), we replace layernorm (Ba et al., 2016) with batchnorm (Ioffe & Szegedy, 2015) before each multilayer perceptron (MLP) block (see Appendix C.1 for a comparison).

**Predictors.** The students contain Transformer predictors, which regress targets given 1) the encoder outputs corresponding to the unmasked portions of the inputs and 2) mask tokens associated with the masked portions. Note that mask tokens are applied to the predictors rather than the encoders (He et al., 2021). This reduces the discrepancy between pre-training and fine-tuning: In both cases, the encoders do not see mask tokens. A predictor that takes representations corresponding to modality $m_1 \in \{v, a\}$ and predicts targets associated with modality $m_2 \in \{v, a\}$ is denoted as $f_p^{m_1 \to m_2}$. For ease, the application of mask tokens to the encoder outputs is absorbed in the notation.

Unlike other works which output global representations (one embedding per sample) and thus use MLPs as predictors (Grill et al., 2020; Chen et al., 2021), we use Transformers to allow modelling temporal dynamics, which we found greatly improves results. The predictors can be lightweight: Indeed, two-block Transformers with hidden size 512 work optimally in our experiments.

**Targets.** The targets are simply the outputs of momentum-based teachers $g^v$ and $g^a$ (Grill et al., 2020; Caron et al., 2021), which are given as input the unmasked video or audio, in order to force the students to predict the missing information. Each teacher is architecturally identical to its student encoder counterpart. Denoting the parameters of the student encoders and teachers as $s^m$ and $t^m$, respectively, at each iteration the following update is performed:

$$t^m \leftarrow \mu t^m + (1 - \mu) s^m, \tag{1}$$

where $m \in \{v, a\}$ specifies the modality and $\mu$ is a momentum parameter following a cosine schedule from 0.999 to 1. A high value of $\mu$ leads to slowly-varying teachers and stable targets. The use of momentum-based teachers obviates the need for handcrafted targets or multi-stage training.

**Prediction tasks.** The auditory modality contains more information relevant to speech than the visual one: Mouth motion is inherently more ambiguous than a speech waveform due to the presence of homophemes (Chung et al., 2017). We propose a loss structure which reflects this asymmetry between the modalities. The audio student predicts the targets from both the video and audio teacher, thus benefiting from the ability of cross-modal learning to induce semantic representations, while at the same time being encouraged to retain information from the auditory input that is absent from the visual one. As a result, two predictors are associated with the audio student, one for each target type. On the other hand, the video student only predicts the auditory targets, which are inevitably of higher quality.

**Losses.** The loss function is the negative cosine similarity (Grill et al., 2020), denoted as sim. Due to the temporal alignment of the inputs, the cosine similarity is applied between pairs of corresponding features and then summed across the time dimension. For the within-modal task (audio-to-audio prediction), the loss is applied only on targets corresponding to masked portions of the input (Devlin et al., 2019). For the cross-modal tasks, the loss is applied on all targets, which we found to work better. Note that predicting the unmasked portions in cross-modal learning is a non-trivial task (unlike in within-modal learning) and can bolster representation learning.

Denoting the set of mask token indices for audio as $M_a$, the audio-to-audio prediction loss and cross-modal losses can be respectively expressed as

$$\mathcal{L}^{a \to a} = - \sum_{n \in M_a} \mathrm{sim}\left(f_p^{a \to a}\left(f_e^a\left(\bar{x}^a\right)\right)_n, \mathrm{sg}\left(g^a\left(x^a\right)_n\right)\right), \tag{2}$$

$$\mathcal{L}^{m_1 \to m_2} = - \sum_{n} \mathrm{sim}\left(f_p^{m_1 \to m_2}\left(f_e^{m_1}\left(\bar{x}^{m_1}\right)\right)_n, \mathrm{sg}\left(g^{m_2}\left(x^{m_2}\right)_n\right)\right), \tag{3}$$

where $m_1, m_2 \in \{v, a\}$, $m_1 \neq m_2$, and sg denotes the "stop-gradient" operation, which indicates that no gradient is passed back to the teacher networks..

**Objectives.**  At each iteration, the objectives for the video and audio students are, respectively,

$$\mathcal{L}_v = \mathcal{L}^{v \to a}, \quad \mathcal{L}_a = \mathcal{L}^{a \to v} + \mathcal{L}^{a \to a}. \tag{4}$$

The teachers are updated via Equation 1.

## 3.2  FINE-TUNING

For fine-tuning, we keep the pre-trained student encoders and discard the rest. We append a linear layer and a Transformer decoder for joint CTC / attention decoding (Watanabe et al., 2017), as in Ma et al. (2021b). Following Ma et al. (2021b), we set the CTC weight to 0.1. We use SentencePiece (Kudo & Richardson, 2018) subword units with a vocabulary size of 1,000 as our targets.

**Self-training.**  Combining pre-training with self-training tends to improve results over using either strategy in isolation (Xu et al., 2021; Shi et al., 2022). To that end, we first fine-tune our pre-trained audio encoder on the labelled data, and then use the model for pseudo-labelling the unlabelled data. The pre-trained video and audio models are then fine-tuned using both the labels and pseudo-labels.

## 4  EXPERIMENTS

### 4.1  SETUP

**Datasets.**  For pre-training, we conduct experiments with LRS3 (Afouras et al., 2018b) (without the labels) as well as a combination of LRS3 and an English-only version of VoxCeleb2 (Chung et al., 2018) curated by Shi et al. (2022), which we refer to as LRS3+Vox2-en. The former features 433 hours of footage and the latter 1,759 hours. For fine-tuning, we use the full LRS3 with the labels as our high-resource labelled data setting, as well as a 30-hour subset (the "trainval" partition) as our low-resource setting. We present results for the LRS2 dataset (Chung et al., 2017) in Appendix B.

**Transformer encoder.**  We show results for two configurations of the Transformer encoders, Base and Large, with 12 and 24 blocks respectively. The hidden size/MLP size/attention heads are 512/2048/8 for Base and 1024/4096/16 for Large, amounting to 41 million (M) and 328 M parameters, respectively. We note that our Base model is around half the size of the one used by Shi et al. (2022) and the Large models are similar in size. Training details are provided in Appendix A.

### 4.2  LOW-RESOURCE LABELLED DATA SETTING

We pre-train our models on LRS3 and/or LRS3+Vox2-en and then fine-tune them on the 30-hour LRS3 subset to evaluate performance when labels are scarce. We reports results in Table 1.

Compared with training from scratch, RAVEn pre-training leads to dramatic performance improvements in all configurations. Notably, increasing the model size *hurts* VSR performance when training from scratch, but *improves* it when using pre-training.

Our Base variant outperforms all related methods on VSR. It surpasses the Base AV-HuBERT model by 4.8% and 5.9% WER when using LRS3 and LRS3+Vox2-en, respectively, despite having roughly half the number of parameters. The Large model provides significant boosts over the Base model (32.5% vs 40.2% WER) when using LRS3+Vox2-en for pre-training, keeping the number of labelled data points fixed. Self-training further improves WER by 7.7%, indicating its complementarity with RAVEn pre-training. Finally, using a language model (see Appendix A.4 for details) leads to a WER of 23.8%, better than a method (Serdyuk et al., 2021) trained on *90,000* hours of non-public data.

On ASR, RAVEn significantly outperforms the audio-only Hubert (Hsu et al., 2021) model, and in all cases is better than or on par with AV-HuBERT. Our best ASR model without self-training achieves 2.7% WER vs AV-HuBERT's 2.9% WER, despite AV-HuBERT using a different pre-training strategy for VSR than ASR. For example, using the same pre-training hyperparameters increases AV-HuBERT's WER for ASR from 3.8% to 4.6% with the Base model (Shi et al., 2022). In contrast,

| Method | Encoder | LM | Unlab hrs | Lab hrs | WER (%) | |
|---|---|---|---|---|---|---|
| | | | | | VSR | ASR |
| *supervised* | | | | | | |
| Afouras et al. (2018a) | Transformer | ✓ | - | 1,519[*] | 58.9 | 8.3 |
| Xu et al. (2020) | RNN | ✗ | - | 590 | 57.8 | 7.2 |
| Shillingford et al. (2019) | RNN | ✓ | - | 3,886[*] | 55.1 | - |
| Ma et al. (2022) | Conformer | ✓ | - | 813 | 34.7 | - |
| Makino et al. (2019) | RNN | ✗ | - | 31,000[*] | 33.6 | 4.8 |
| Prajwal et al. (2022) | Transformer | ✓ | - | 2,676[*] | 30.7 | - |
| Serdyuk et al. (2021) | Transformer | ✗ | - | 90,000[*] | 25.9 | 2.3 |
| Serdyuk et al. (2022) | Conformer | ✗ | - | 90,000[*] | **19.3** | **1.6** |
| *from scratch* | | | | | | |
| Base model | Transformer | ✗ | - | 30 | 93.4 | 18.5 |
| Large model | Transformer | ✗ | - | 30 | 95.5 | 9.9 |
| *self-supervised* | | | | | | |
| **Base models, less pre-training data** | | | | | | |
| Ma et al. (2021a) | Transformer | ✗ | 433 | 30 | 71.9[†] | - |
| Zhang et al. (2022) | Transformer | ✗ | 433 | 30 | 67.8 | 10.9 |
| Hsu et al. (2021) | Transformer | ✗ | 433 | 30 | - | 5.4 |
| Shi et al. (2022) | Transformer | ✗ | 433 | 30 | 51.8 | 4.9 |
| RAVEn | Transformer | ✗ | 433 | 30 | **47.0** | **4.7** |
| **Base models, more pre-training data** | | | | | | |
| Hsu et al. (2021) | Transformer | ✗ | 1,759 | 30 | - | 5.0 |
| Shi et al. (2022) | Transformer | ✗ | 1,759 | 30 | 46.1 | 4.6[‡] / **3.8** |
| RAVEn | Transformer | ✗ | 1,759 | 30 | **40.2** | **3.8** |
| **Large models, more pre-training data** | | | | | | |
| Hsu et al. (2021) | Transformer | ✗ | 1,759 | 30 | - | 3.2 |
| Shi et al. (2022) | Transformer | ✗ | 1,759 | 30 | 32.5 | 2.9 |
| Shi et al. (2022) w/ self-training | Transformer | ✗ | 1,759 | 30 | 28.6 | - |
| RAVEn | Transformer | ✗ | 1,759 | 30 | 32.5 | 2.7 |
| RAVEn w/ self-training | Transformer | ✗ | 1,759 | 30 | 24.8 | 2.3 |
| RAVEn w/ self-training | Transformer | ✓ | 1,759 | 30 | **23.8** | **1.9** |

Table 1: **LRS3 low-resource setting**. We report results on the test set when fine-tuning on 30 hours of LRS3 with different model sizes and number of unlabelled data hours (Unlab hrs). LM denotes whether or not a language model was used during decoding. We also provide baselines for training our models from scratch (without our pre-training) and results from fully-supervised methods trained on more labelled data hours (Lab hrs) for reference. [*]Includes non-publicly-available data. [†]Result taken from Shi et al. (2022). [‡]Result with the same pre-training strategy for VSR and ASR.

the video and audio encoders we use for fine-tuning are the result of a *single* pre-training phase, where they were jointly learned.

All in all, our results suggest that transcribed data, which are costly to obtain, can be largely substituted with raw unlabelled audiovisual data.

### 4.3 HIGH-RESOURCE LABELLED DATA SETTING

Table 2 reports results when fine-tuning on the full 433 hours of LRS3. Despite increasing the labelled data, training the model *from scratch* still leads to poor performance, as in the low-resource setting. This is likely related to the long utterances in the LRS3 dataset, which may cause optimisation difficulties (Ma et al., 2022). A potential remedy is to employ curriculum learning, as proposed by Ma et al. (2022), by training the network in multiple stages with increasingly longer sequences. We observe that this strategy indeed reduces WER from 87.3% to 39.8% for the Base model. Even so, pre-training with the same data used for fine-tuning leads to a better WER of 39.1%, *without* requiring a curriculum learning procedure. This suggests that self-supervised pre-training can aid optimisability. The impact of pre-training is even more pronounced with the Large model.

| Method | Encoder | LM | Unlab hrs | Lab hrs | WER (%) | |
|---|---|---|---|---|---|---|
| | | | | | VSR | ASR |
| *semi-supervised (using external models for pseudo-labelling)* | | | | | | |
| Afouras et al. (2020) | CNN | ✓ | 344 | 433 | 59.8 | - |
| Ma et al. (2022) | Conformer | ✓ | 641 | 818 | **31.5** | - |
| *from scratch* | | | | | | |
| Base model | Transformer | ✗ | - | 433 | 87.3 | 2.5 |
| Base model w/ curriculum | Transformer | ✗ | - | 433 | 39.8 | - |
| Large model | Transformer | ✗ | - | 433 | 85.8 | **2.2** |
| Large model w/ curriculum | Transformer | ✗ | - | 433 | **38.8** | - |
| *self-supervised* | | | | | | |
| **Base models, less pre-training data** | | | | | | |
| Shi et al. (2022) | Transformer | ✗ | 433 | 433 | 44.0 | - |
| RAVEn | Transformer | ✗ | 433 | 433 | **39.1** | **2.2** |
| **Base models, more pre-training data** | | | | | | |
| Ma et al. (2021a) | Transformer | ✗ | 1,759 | 433 | 49.6[†] | - |
| Hsu et al. (2021) | Transformer | ✗ | 1,759 | 433 | - | 2.4 |
| Shi et al. (2022) | Transformer | ✗ | 1,759 | 433 | 34.8 | 2.0 |
| RAVEn | Transformer | ✗ | 1,759 | 433 | **33.1** | **1.9** |
| **Large models, more pre-training data** | | | | | | |
| Hsu et al. (2021) | Transformer | ✗ | 1,759 | 433 | - | 1.5 |
| Shi et al. (2022) | Transformer | ✗ | 1,759 | 433 | 28.6 | **1.3** |
| Shi et al. (2022) w/ self-training | Transformer | ✗ | 1,759 | 433 | 26.9 | - |
| RAVEn | Transformer | ✗ | 1,759 | 433 | 27.8 | 1.4 |
| RAVEn w/ self-training | Transformer | ✗ | 1,759 | 433 | 24.4 | 1.4 |
| RAVEn w/ self-training | Transformer | ✓ | 1,759 | 433 | **23.1** | 1.4 |

Table 2: **LRS3 high-resource setting**. We report results on the test set with different model sizes and number of unlabelled data hours (Unlab hrs). Lab hrs denotes the number of labelled hours, and LM denotes whether or not a language model was used during decoding. We provide baselines for training our models from scratch (without our pre-training). [†]Result taken from Shi et al. (2022).

RAVEn outperforms AV-HuBERT under all configurations on VSR. Our best result is 23.1%, achieved using self-training and a language model. We note that unlike methods that use external ASR models (Afouras et al., 2020; Ma et al., 2022) for pseudo-labelling, we do not require extra data for the self-training phase.

We are on par with the state-of-the-art for ASR in the high-resource setting, achieving a WER of 1.4% with the Large model. This is despite using raw audio as input (rather than spectrograms which Shi et al. (2022) use). We notice that additionally including self-training and a language model does not reduce the WER, suggesting that the ASR performance may have saturated in our environment.

## 4.4 PRE-TRAINING ABLATIONS

Ablations are performed with our Base model in the low-resource setting with LRS3 pre-training using the validation set from Shi et al. (2022) (as there is no official development set). For more ablations, see Appendix C.

**Prediction tasks.** We study in Table 3 the impact of cross- and within-modal prediction on the learned representations. Figure 2 illustrates the various prediction tasks we consider. We observe the following. The *within-modal* variant performs decently for ASR but poorly for VSR, which can be explained by the assumption that audio contains more speech information than video. The *cross-modal* variant performs better than within-modal for both modalities. Since lip movements and the corresponding waveform are correlated in terms of lexical content, the representations resulting from cross-modal prediction are expected to capture rich semantic information. At the same time, they are likely to be, to a large extent, invariant to factors unshared between the two modalities,

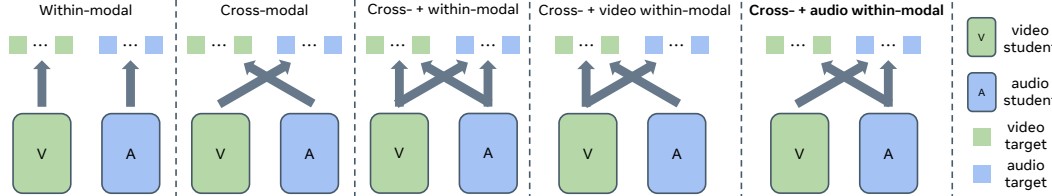

Figure 2: **Prediction tasks**. We consider different choices for our prediction tasks, combining cross- and within-modal losses. We find that applying both cross- and within-modal losses for the audio student, and only a cross-modal loss for the video student works best.

| Setting | Prediction tasks | | | | WER (%) | |
|---|---|---|---|---|---|---|
| | $V \to V$ | $A \to A$ | $V \to A$ | $A \to V$ | VSR | ASR |
| Within-modal | ✓ | ✓ | ✗ | ✗ | 92.7 | 15.5 |
| Cross-modal | ✗ | ✗ | ✓ | ✓ | 40.8 | 14.0 |
| Cross- + within-modal | ✓ | ✓ | ✓ | ✓ | 49.0 | 14.0 |
| Cross- + video within-modal | ✓ | ✗ | ✓ | ✓ | 55.3 | 16.4 |
| Cross- + audio within-modal | ✗ | ✓ | ✓ | ✓ | **32.9** | **12.2** |

Table 3: **Prediction task** ablations under the LRS3 low-resource setting using our Base model. V $\to$ A means that the video student predicts the audio teacher representations. Each prediction loss is associated with a separate predictor.

such as visual or auditory noise, benefiting generalisation. Combining *cross- with within-modal* learning hurts VSR relative to only cross-modal prediction. However, cross-modal with within-modal learning *only for audio* achieves the best results. We hypothesise that predicting the auditory targets forces the auditory stream to keep relevant information absent from the visual stream. We note that removing the video-to-video prediction (from row 3 to row 5 in Table 3) also improves the audio student, as the quality of the visual targets improves. Finally, cross-modal with within-modal learning *only for video* does not perform well, validating that the asymmetry only works one way.

**Masking sampling.** Table 4a studies the effect of different masking strengths by varying the mask length and probability that a given index is chosen as the mask start. We observe that a probability of 0.2 and a length of three video frames (corresponding to 1,920 audio samples) works well. Less masking allows the student to focus less on context and degrades both VSR and ASR. Interestingly, although more masking hurts VSR, it helps ASR, suggesting that an asymmetric masking strategy w.r.t. the two modalities may further improve results. We leave this exploration to future work.

Our masking is less aggressive than what was found to be optimal in related self-supervised image and action recognition literature (where 75% or even 90% of the input is masked) (He et al., 2021; Tong et al., 2022). We hypothesise that mouth movements are fast-varying and do not contain much temporal redundancy, and thus such strong masking makes our pretext task overly difficult.

| Prob | Length | WER (%) | |
|---|---|---|---|
| | | VSR | ASR |
| 0.1 | 1 | 41.1 | 15.6 |
| 0.2 | 3 | **32.9** | 12.2 |
| 0.4 | 5 | 35.7 | **11.7** |

(a) **Masking strength** in terms of starting index probability and mask length.

| Position | WER (%) | |
|---|---|---|
| | VSR | ASR |
| Encoder | 43.1 | 14.0 |
| Predictor | **32.9** | **12.2** |

(b) **Mask token position.** Applying the mask tokens in the predictor works better than in the encoder.

| Loss | WER (%) | |
|---|---|---|
| | VSR | ASR |
| Masked | 44.3 | 12.6 |
| All | **32.9** | **12.2** |

(c) **Cross-modal loss.** Applying the cross-modal loss for all inputs is superior to applying it only for masked inputs.

Table 4: **Masking** ablations under the LRS3 low-resource setting using our Base model.

| Predictor type | Params | WER (%) | |
|---|---|---|---|
| | | VSR | ASR |
| None | 0 | coll.[‡] | coll.[‡] |
| Linear | 0.3M | 64.4 | 17.7 |
| MLP | 4.2M | 62.6 | 19.0 |
| Transformer 1 block | 3.9M | 46.7 | 14.1 |
| Transformer 2 blocks | 7.4M | **32.9** | **12.2** |
| Transformer 4 blocks | 14.2M | 46.2 | 13.4 |

(a) **Predictor type.** A lightweight Transformer predictor works better than a linear layer or 2-layer MLP design. [‡]Denotes representation collapse.

| Att dim | MLP dim | WER (%) | |
|---|---|---|---|
| | | VSR | ASR |
| 256 | 1024 | 44.0 | 13.3 |
| 512 | 2048 | **32.9** | **12.2** |
| 1024 | 4096 | 39.1 | 12.3 |

(b) **Predictor width.** A moderate predictor width works optimally.

Table 5: **Predictor capacity** ablations under the LRS3 low-resource setting using our Base model.

**Mask token position.** We investigate in Table 4b the optimal position of the mask tokens. It is clear that better performance is achieved when applying them in the predictors rather than the encoders. Forgoing the use of mask tokens in the encoders leads to no input discrepancy between pre-training and fine-tuning, since no mask tokens are used during fine-tuning.

**Mask loss.** It is common to apply the loss only on masked inputs for *within-modal* losses (Hsu et al., 2021; He et al., 2021), since predicting targets corresponding to unmasked inputs may be trivial. However, this is not the case for cross-modal prediction, where the targets are not related to the inputs in an obvious way. Indeed, Table 4c shows that applying the loss for both masked and unmasked inputs outperforms applying it only for masked inputs.

**Predictor design.** Table 5a shows the effect of different predictor types. We note that using no predictors at all leads to representation collapse, where all outputs are constant (Grill et al., 2020). We compare linear layers and 2-layer MLPs (applied independently to each encoder output) with Transformers of varying capacity. Following Grill et al. (2020), the MLPs have hidden dimension of 4096 with batchnorm followed by ReLU (Nair & Hinton, 2010) after the hidden layer.

We find that a Transformer predictor works significantly better, even when the number of parameters in the (1-block) Transformer and the 2-layer MLP is similar. Interestingly, the ability for the predictors to model temporal dependencies seems to be crucial to our representation learning phase.

A two-block Transformer works optimally (Table 5a). A shallower Transformer likely results in representations too specialised to the pretext task. A deeper one might place a lot of the burden on the predictors, leaving the encoder representations too general. Similar conclusions can be drawn regarding the Transformer width (see Table 5b). Overall, the lightweight predictor design means that using three predictors (one on the video student side and two on the audio) has relatively little effect on the total computational cost.

## 5  CONCLUSION

We presented RAVEn, a single-stage method that jointly learns visual and auditory speech representations entirely from raw data. It employs momentum encoders to generate targets, which, given masked inputs, are predicted by Transformer encoders and predictors. Especially salient to the quality of the representations are appropriate masking, lightweight Transformer predictors, and an asymmetric loss structure w.r.t. the visual and auditory modalities. RAVEn achieves strong performance under many settings without requiring multiple pre-training stages, handcrafted audio targets, or separate pre-training strategies for VSR and ASR.

As future work, it would be interesting to examine the effect of sharing weights between the visual and auditory encoders as a way of reducing the memory requirements for pre-training. We would also like to apply RAVEn to other tasks related to speech. We hope our study inspires future research extending beyond speech recognition.

ETHICS STATEMENT

Despite numerous positive applications, our method can also be misused. For example, lipreading technologies can be employed for surveillance, compromising the public's privacy and trust. This problem is likely to be exacerbated as the quality of CCTV cameras improves over time. Appropriate government regulations will need to be put in place to limit such concerns. Another issue relates to the potential biases embedded in the datasets used. Such biases may have to do with gender, age, or ethnic backgrounds. A specific group being under-represented in the data would likely result in reduced model performance on samples belonging to said group. Making sure that the models are trained on balanced data, or using other bias-reduction techniques, can address such issues.

Our work used datasets that were made public for research purposes, i.e., LRS2, LRS3, and Vox-Celeb2. In particular, we used the cropped face .mp4 videos that were available on the official dataset webpages and which complied with the following licenses: Creative Commons BY-NC-ND 4.0 license and Creative Commons Attribution 4.0 International License, TED terms of use, and BBC's terms of use.

REPRODUCIBILITY

To ensure reproducibility, we provide as many implementation details as possible in the main paper as well as tables showing the hyperparameter values in the appendix. Moreover, we plan on making the code and pre-trained models publicly available.

ACKNOWLEDGEMENTS

Only non-Meta co-authors downloaded, accessed, and used the LRS2 dataset. Only non-Meta authors conducted any of the dataset pre-processing (no dataset pre-processing took place on Meta's servers or facilities).

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

| Hyperparameter | Value |
| --- | --- |
| Training epochs | 150 |
| Warmup epochs | 40 (LRS3), 30 (LRS3+Vox2-en) |
| Optimiser | AdamW |
| Learning rate | 3e-3 (Base), 2e-3 (Large) |
| Optimiser $(\beta_1, \beta_2)$ | (0.9, 0.999) |
| Weight decay | 0.04 |
| Learning rate schedule | Cosine decay |
| Drop path | 0.05 (Base), 0.1 (Large) |
| Gradient clip | 5.0 |
| Video augmentations | RandomCrop + HorizontalFlip |

Table 6: **Pre-training settings.**

# A DATASET / IMPLEMENTATION DETAILS

## A.1 DATASETS

**LRS3.** LRS3 (Afouras et al., 2018b) is the largest publicly available transcribed audio-visual dataset for continuous speech recognition. It consists of around 430 hours of spoken sentences from TED talks, with a vocabulary of more than 50,000 words uttered by thousands of speakers. The test set contains around 1 hour of utterances with speakers separate from those in the training set.

**LRS2.** The 223-hour LRS2 dataset (Chung et al., 2017), collected from BBC programmes, is the second-largest publicly available transcribed audio-visual dataset for continuous speech recognition. As LRS3, it contains an unconstrained vocabulary and thousands of diverse speakers.

**VoxCeleb2.** VoxCeleb2 (Chung et al., 2018) is a non-transcribed dataset containing YouTube-downloaded videos. It consists of around 2,500 hours of utterances with over 6,000 speakers. Since VoxCeleb2 is multi-lingual, as mentioned in the main text, we use an English-only version curated by Shi et al. (2022), amounting to 1,759 hours.

## A.2 DATASET PRE-PROCESSING

We follow common practices in the literature for dataset pre-processing (Ma et al., 2022; Shi et al., 2022; Martinez et al., 2020). We crop a $96 \times 96$ region centred around the mouth and transform to grayscale. Raw audio is used without any pre-processing nor normalisation. Utterances longer than 24 seconds are split into smaller constituents.

## A.3 PRE-TRAINING

Table 6 provides the default setting for pre-training. We use the AdamW (Loshchilov & Hutter, 2019) optimiser with linear learning rate warmup (Goyal et al., 2017) and a cosine decay schedule (Loshchilov & Hutter, 2017). During training, we apply random spatial cropping of size ($88 \times 88$) followed by horizontal flipping with probability 0.5. These augmentations are applied in a time-consistent manner across the video clips to maintain temporal coherence. We do not use any augmentations for the raw audio. We also use stochastic depth (Huang et al., 2016) for regularisation, as well as LayerScale (Touvron et al., 2021) with coefficient 0.1. We train the Base model with 32 A100 GPUs, and the Large model with 128. It takes around 15 minutes / 1 hour per epoch to pre-train our Base models on LRS3 / LRS3+Vox2-en, and 2 hours per epoch for our Large model on LRS3+Vox2-en.

Our batching process is as follows: The samples are sorted based on their length to minimise zero-padding, and then samples with the same length are randomly shuffled at the beginning of each training epoch. Each batch contains a maximum of 96 / 36 seconds of footage for Base / Large.

| Hyperparameter | Value |
|---|---|
| Training epochs | 75 (HR), 50 (LR) |
| Warmup epochs | 20 |
| Optimiser | AdamW |
| Learning rate encoder | 2e-3 |
| Learning rate decoder | 6e-3 |
| Optimiser $(\beta_1, \beta_2)$ | (0.9, 0.98) |
| Weight decay | 0.04 (HR), 0.1 (LR) |
| Learning rate schedule | Cosine decay |
| Layer-wise learning rate decay | 0.75 (HR), 0.5 (LR) |
| Minimum learning rate | 1e-5 |
| Drop path | 0.2 |
| Gradient clip | 5.0 |
| Video augmentations | RandomCrop + HorizontalFlip + TimeMask |
| Audio augmentations | TimeMask |
| Decoder blocks | 6 (Base), 9 (Large) |
| Decoder hidden size | 256 (LR), 512 (HR, Base), 1024 (HR, Large) |
| Decoder MLP size | 1024 (LR), 2048 (HR, Base), 4096 (HR, Large) |
| Decoder attention heads | 4 (LR), 8 (HR, Base), 16 (HR, Large) |

Table 7: **Fine-tuning settings.** HR denotes high-resource labelled data setting and LR low-resource.

### A.4 Fine-tuning / decoding

The protocol for fine-tuning is similar to that for pre-training with some exceptions (see Table 7). We use a higher learning rate for the decoder than for the pre-trained encoder, since the decoder is randomly-initialised. We also employ layer-wise learning rate decay (Clark et al., 2020), which we found to reduce overfitting. In addition to the augmentations from the pre-training stage, we apply time masking (Ma et al., 2022) for both video and audio clips. Specifically, for each second of the sample, we use zero-masking with a duration that is uniformly sampled between 0 and 0.4 seconds. Fine-tuning Base (for both VSR and ASR) takes around 30 seconds per epoch in the low-resource setting and 7 minutes per epoch in the high-resource setting. Fine-tuning Large takes approximately 1 minute and 20 minutes per epoch in the low- and high-resource setting, respectively.

**CTC/attention decoding.** We use joint CTC/attention decoding to map the input sequence $x$ of length $T$ to a target sequence $y = (y_1, \ldots, y_L)$ of size $L$. The CTC loss during fine-tuning is given by

$$\mathcal{L}_{\text{ctc}} = -\log \sum_{A \in A_{x,y}} \left( \prod_{t=1}^{T} p_t (a_t | x) \right), \tag{5}$$

where $A_{x,y}$ is the set of valid alignments, $A$ is one such alignment, and $a_t$ is the token at time-step $t$.

The attention loss, computed using the Transformer decoder outputs, can be expressed as

$$\mathcal{L}_{\text{att}} = -\sum_{l=1}^{L} \log p (y_l | y_1, \ldots, y_{l-1}, x). \tag{6}$$

The final loss during fine-tuning is given by $\mathcal{L} = \alpha \mathcal{L}_{\text{ctc}} + (1 - \alpha) \mathcal{L}_{\text{att}}$, where $\alpha = 0.1$.

We use the ESPnet framework (Watanabe et al., 2018) for decoding. We set the beam size to 40. The final score used to choose the most likely sequence is given by $\mathcal{S} = \lambda \mathcal{S}_{\text{ctc}} + (1-\lambda) \mathcal{S}_{\text{att}} + \beta \mathcal{S}_{LM}$, where $\mathcal{S}_{\text{ctc}}$ and $\mathcal{S}_{\text{att}}$ denote the scores from the CTC and attention branches, respectively, and $\lambda = 0.1$. $\mathcal{S}_{LM}$ is an optional score from the language model, incorporated through shallow fusion (Watanabe et al., 2017). When using a language model, $\beta$ is chosen from $\{0.1, 0.2, 0.3, 0.4\}$ using the validation set.

Inference on one A100 GPU (without batching) takes around 3 seconds to decode 10 seconds of footage for Base and 5 seconds for Large.

| stage | filters | output size |
|---|---|---|
| conv$_1$ | $5 \times 7 \times 7$, 64, stride $1 \times 2 \times 2$ | $T \times 44 \times 44$ |
| pool$_1$ | max, $1 \times 3 \times 3$, stride $1 \times 2 \times 2$ | $T \times 22 \times 22$ |
| res$_1$ | $\begin{bmatrix} 3 \times 3, 64 \\ 3 \times 3, 64 \end{bmatrix} \times 2$ | $T \times 22 \times 22$ |
| res$_2$ | $\begin{bmatrix} 3 \times 3, 128 \\ 3 \times 3, 128 \end{bmatrix} \times 2$ | $T \times 11 \times 11$ |
| res$_3$ | $\begin{bmatrix} 3 \times 3, 256 \\ 3 \times 3, 256 \end{bmatrix} \times 2$ | $T \times 6 \times 6$ |
| res$_4$ | $\begin{bmatrix} 3 \times 3, 512 \\ 3 \times 3, 512 \end{bmatrix} \times 2$ | $T \times 3 \times 3$ |
| pool$_2$ | global spatial average pool | $T \times 1 \times 1$ |

(a) **Visual feature extractor.**

| stage | filters | output size |
|---|---|---|
| conv$_1$ | 80, 64, stride 4 | $160T$ |
| res$_1$ | $\begin{bmatrix} 3, 64 \\ 3, 64 \end{bmatrix} \times 2$ | $160T$ |
| res$_2$ | $\begin{bmatrix} 3, 128 \\ 3, 128 \end{bmatrix} \times 2$ | $80T$ |
| res$_3$ | $\begin{bmatrix} 3, 256 \\ 3, 256 \end{bmatrix} \times 2$ | $40T$ |
| res$_4$ | $\begin{bmatrix} 3, 512 \\ 3, 512 \end{bmatrix} \times 2$ | $20T$ |
| pool$_2$ | average pool, stride 20 | $T$ |

(b) **Auditory feature extractor.**

Table 8: **Feature extractors**. We provide details on the convolutional architectures used as the feature extractors for the visual and auditory modalities. Note that the output sizes for both networks match.

**Language model.** We use a 16-block Transformer-based language model, as proposed by Irie et al. (2019). The hidden size/MLP size/attention heads are 512/2048/8. The language model is trained on the combination of the following datasets: Librispeech (Panayotov et al., 2015), LRS2/3, TED-LIUM 3 (Hernandez et al., 2018), VoxForge and Common Voice (Ardila et al., 2020). The total number of characters is 166 million.

## A.5 FEATURE EXTRACTORS

The details of the visual and auditory convolutional feature extractors are provided in Tables 8a and 8b, respectively.

## B LRS2 EXPERIMENTS

We report results on the test set of the LRS2 dataset (Chung et al., 2017) in Table 9. After pre-training on the LRS3 or the LRS3+Vox2-en datasets, we fine-tune on the "pre-training" and "training" sets of LRS2. We observe similar trends as in the LRS3 experiments (Table 2), namely that performance is benefited from large models, large unlabelled datasets, and self-training. We significantly outperform all other methods, including one (Pan et al., 2022) pre-trained on 60,000 hours of audio data.

## C MORE ABLATIONS

### C.1 MORE PRE-TRAINING ABLATIONS

**Momentum parameter.** Table 10a shows the effect of varying the momentum parameter for updating the teacher networks. We show results at three coarse levels: 0 (teacher is a copy of the student at each iteration), 0.999 (with a cosine schedule to 1), and 1 (teacher does not get updated during training). We see that using a momentum value of 0 leads to representation collapse. At the other extreme, a value of 1 does not allow the teacher targets to improve during training, leading to poor representations. A slowly-evolving momentum encoder is most effective.

**Pre-MLP normalisation.** As mentioned in the main text, we use batchnorm before each MLP module rather than layernorm, a choice inspired by Chen et al. (2021). Table 10b shows that our method still works with layernorm, but lags behind batchnorm. This is an interesting observation worthy of future exploration. A preliminary hypothesis is that batchnorm may improve the

| Method | Encoder | LM | Unlab hours | Lab hours | WER (%) | |
|---|---|---|---|---|---|---|
| | | | | | VSR | ASR |
| *supervised* | | | | | | |
| Chung et al. (2017) | LSTM | ✓ | - | 223 | 70.4 | - |
| Petridis et al. (2018) | LSTM | ✓ | - | 380 | 63.5 | 8.3 |
| Ren et al. (2021) | Transformer | ✗ | - | 818 | 49.2 | - |
| Yu et al. (2020) | CNN | ✓ | - | 223 | 48.9 | 6.7 |
| Ma et al. (2021b) | Conformer | ✓ | - | 380 | 37.9 | 3.9 |
| Prajwal et al. (2022) | Transformer | ✓ | - | 698 | 28.9 | - |
| Ma et al. (2022) | Conformer | ✓ | - | 818 | 27.3 | - |
| Prajwal et al. (2022) | Transformer | ✓ | - | 2,676* | **22.6** | - |
| *semi-supervised (using external models for pseudo-labelling)* | | | | | | |
| Afouras et al. (2020) | CNN | ✓ | 777 | 223 | 51.3 | - |
| Ma et al. (2022) | Conformer | ✓ | 641 | 818 | **25.5** | - |
| *self-supervised* | | | | | | |
| Pan et al. (2022) | Transformer | ✗ | 60,000 | 223 | 43.2 | 2.7 |
| Ma et al. (2021a) | Conformer | ✓ | 433 | 223 | 38.8 | - |
| RAVEn (Base) | Transformer | ✗ | 433 | 223 | 32.1 | 3.9 |
| RAVEn (Large) | Transformer | ✗ | 1,759 | 223 | 23.2 | 2.5 |
| RAVEn (Large) w/ self-training | Transformer | ✗ | 1,759 | 223 | 19.3 | **2.3** |
| RAVEn (Large) w/ self-training | Transformer | ✓ | 1,759 | 223 | **17.9** | **2.3** |

Table 9: **LRS2 results.** We report results on the test set with different model sizes and number of unlabelled data hours (Unlab hours). Lab hours denotes the number of labelled hours, and LM denotes whether or not a language model was used during decoding. *Includes non-publicly available data.

| Momentum $\mu$ | WER (%) | |
|---|---|---|
| | VSR | ASR |
| 0 | coll.‡ | coll.‡ |
| 0.999† | **32.9** | **12.2** |
| 1 | 75.0 | 74.0 |

(a) **Momentum parameter.** It is important for the momentum encoders to slowly evolve during training. †Follows a cosine schedule with 0.999 as the starting value. ‡Denotes representation collapse.

| Norm | WER (%) | |
|---|---|---|
| | VSR | ASR |
| BN | **32.9** | **12.2** |
| LN | 39.0 | 12.7 |

(b) **Pre-MLP normalisation.** Using batchnorm (BN) before the MLPs in the encoder works better than using layernorm (LN).

Table 10: **More pre-training** ablations under the LRS3 low-resource setting using our Base model.

| Encoder LR | Decoder LR | WER (%) | |
|---|---|---|---|
| | | VSR | ASR |
| *same LR* | | | |
| $1 \times 10^{-3}$ | $1 \times 10^{-3}$ | 36.1 | 13.1 |
| $3 \times 10^{-3}$ | $3 \times 10^{-3}$ | 34.0 | **12.2** |
| $5 \times 10^{-3}$ | $5 \times 10^{-3}$ | 34.8 | **12.2** |
| *different LR* | | | |
| $1 \times 10^{-3}$ | $5 \times 10^{-3}$ | **32.9** | **12.2** |

(a) **Learning rates (LR) for encoder and decoder.** Using a larger learning rate for the decoder than the encoder is beneficial for VSR.

| Learning rate decay | WER (%) | |
|---|---|---|
| | VSR | ASR |
| 1.0 | 42.4 | 12.5 |
| 0.75 | 33.1 | **11.7** |
| 0.50 | **32.9** | 12.2 |
| 0.25 | 33.0 | 12.5 |

(b) **Learning rate decay.** Reducing the learning rate as the encoder depth decreases improves performance.

| Tokens | LM | WER (%) | |
|---|---|---|---|
| | | VSR | ASR |
| Character | ✗ | 35.8 | 13.6 |
| Character | ✓ | 32.7 | 11.1 |
| Subword | ✗ | 32.9 | 12.2 |
| Subword | ✓ | **30.4** | **10.5** |

(c) **Tokenisation.** Using subword (SentencePiece) units as targets is better than using characters. A language model (LM) also helps.

Table 11: **Fine-tuning** ablations under the LRS3 low-resource setting using our Base model.

conditioning of the networks at initialisation, leading to better targets at the beginning of training (Richemond et al., 2020).

## C.2 FINE-TUNING ABLATIONS

**Learning rates.** Table 11a studies the effect of using different learning rates for the encoder and decoder. We find that it is beneficial for VSR to use a higher learning rate for the decoder than the encoder, likely because the encoder is pre-trained, whereas the decoder is randomly-initialised and thus requires a larger learning rate to search for a more effective local optimum.

**Learning rate decay.** Table 11b shows the influence of decaying the encoder's learning rate as the depth of the model decreases. The learning rate at block $b$, $r_b$, is given by $r_b = r_B d^{B-b}$, where $B$ is the index of the last block and $d$ is the learning rate decay (Clark et al., 2020). A decay less than 1 works well for both VSR and ASR, suggesting that during fine-tuning it is useful to employ larger learning rates for the deeper layers, which are more task-specific.

**Tokenisation.** In Table 11c, we compare the use of characters with SentencePiece (Kudo & Richardson, 2018) subword units (vocabulary size of 1,000) as our target tokens. We find that using subword units leads to superior results than character units. This may be explained by the language priors embedded in subword units, which facilitate speech recognition.

## C.3 AUDIO-VISUAL FINE-TUNING

It is possible to use the learned auditory and visual representations for audio-visual speech recognition. To that end, following Ma et al. (2021b), we concatenate the outputs of the two encoders, feed the resulting embeddings to a 2-layer MLP module with hidden size 4096 and batchnorm, and then fine-tune for speech recognition, as described in Section 3.2. We initialise the video and audio encoders with weights obtained from uni-modal fine-tuning and randomly initialise the rest. We fine-tune for 30 epochs with the hyperparameters used for audio-only fine-tuning, shown in Table 7.

Table 12 shows results in the low- and high-resource settings for the Large model with LRS3+Vox2-en pre-training. Audio-visual training gives marginal improvements (if any) compared to audio-only

| Fine-tune setting | Resource | WER (%) |
|---|---|---|
| Audio-only | Low | 2.6 |
| Audio-visual | Low | 2.5 |
| Audio-only | High | 1.4 |
| Audio-visual | High | 1.4 |

Table 12: **Audio vs audio-visual learning on LRS3 test set..** Including the visual modality for fine-tuning has very little influence in clean conditions.

| Model | Transcription |
|---|---|
| Groundtruth | So what do you think happens to these gals |
| VSR, low-resource, base, LRS3 | So what did the thing I would see this talk |
| VSR, high-resource, large, LRS3+Vox2-en | So what do you think happens this is an accident |
| ASR, low-resource, base, LRS3 | So what do you think happens to these look outs |
| ASR, high-resource, large, LRS3+Vox2-en | So what do you think happens to these gals |
| Groundtruth | project really is to find photographs that were taken before something |
| VSR, low-resource, base, LRS3 | projects really has to find photographs on which I can before something |
| VSR, high-resource, large, LRS3+Vox2-en | process really is to find photographs that were taken before something |
| ASR, low-resource, base, LRS3 | project really is to find photographs that we're taking before something |
| ASR, high-resource, large, LRS3+Vox2-en | project really is to find photographs that were taken before something |
| Groundtruth | Why are we embedded in social networks |
| VSR, low-resource, base, LRS3 | Why are we admitted social networks |
| VSR, high-resource, large, LRS3+Vox2-en | Why are we embedded in social networks |
| ASR, low-resource, base, LRS3 | Why are we in better social networks |
| ASR, high-resource, large, LRS3+Vox2-en | Why are we embedded in social networks |
| Groundtruth | They won the game |
| VSR, low-resource, base, LRS3 | Because I wasn't happy |
| VSR, high-resource, large, LRS3+Vox2-en | They want to come |
| ASR, low-resource, base, LRS3 | They want the game |
| ASR, high-resource, large, LRS3+Vox2-en | They won the game |
| Groundtruth | Has it gotten better |
| VSR, low-resource, base, LRS3 | It's not better |
| VSR, high-resource, large, LRS3+Vox2-en | It's gotten better |
| ASR, low-resource, base, LRS3 | Is it gotten better |
| ASR, high-resource, large, LRS3+Vox2-en | Has it gotten better |

Table 13: **Transcription errors**.

when the audio is clean (i.e., not noisy), consistent with findings by Ma et al. (2021b). Investigating the impact of audio-visual training and testing on audio corrupted with various noise types, is outside the scope of this work, but is interesting to take up in future work.

## D    ANALYSIS OF TRANSCRIPTION ERRORS

We provide examples of transcription errors in Table 13. We consider the Base VSR and ASR models fine-tuned in the low-resource setting (47.0% and 4.7% WER, respectively) and the Large VSR and ASR models fine-tuned in the high-resource setting with self-training and a language model (23.1% and 1.4% WER, respectively). Although the worst model sometimes makes surprising errors (*e.g.*, "They won the game" → "Because I wasn't happy"), most often the errors are related to words that are phonetically similar (*e.g.*, "were taken" → "we're taking", "embedded" → "in better", "won" → "want"). As expected, the quality of the transcriptions is higher for ASR than VSR, and it improves as we increase the model size and number of unlabelled data points.

## E    RAVEN FOR VIDEO-TO-SPEECH SYNTHESIS

We evaluate here whether RAVEn pre-training can benefit tasks other than speech recognition by considering video-to-speech synthesis, which aims to output the speech waveform given the corresponding silent video of lip movements. We follow the protocol of SVTS (Mira et al., 2022), the

| Method | Encoder | Training dataset | PESQ | STOI | ESTOI |
|--------|---------|------------------|------|------|-------|
| Mira et al. (2022) | Conformer | LRS3 | 1.25 | 0.51 | 0.27 |
| Mira et al. (2022) | Conformer | LRS3+Vox2-en | 1.26 | 0.53 | 0.31 |
| SVTS from scratch | Transformer | LRS3+Vox2-en | 1.26 | 0.53 | 0.30 |
| SVTS w/ RAVEn | Transformer | LRS3+Vox2-en | **1.30** | **0.56** | **0.36** |

Table 14: **Video-to-speech synthesis on LRS3 test set.** Using RAVEn pre-training for the video encoder provides significant boosts in performance.

current state-of-the-art method, to perform video-to-speech on the LRS3 test set after training on LRS3+Vox2-en. We study the effect of initialising the video encoder with the weights from our Large pre-trained model. We use the same hyperparameters as Mira et al. (2022), except that we use a learning rate of 2e-4 and train only for 30 epochs (rather than 150). We train the "from scratch" baseline with a learning rate of 7e-4 for 50 epochs, as training longer resulted in overfitting.

Table 14 reports performance based on PESQ (Rix et al., 2001), which measures the perceptual quality of the generated samples, as well as STOI and ESTOI (Taal et al., 2011), which measure their intelligiblity. The results show that using RAVEn pre-training results in significant improvements in performance, compared with training from scratch, across all metrics. In fact, the performance boosts due to RAVEn pre-training (0.04 / 0.03 / 0.06 for PESQ / STOI / ESTOI) are larger than those observed by Mira et al. (2022) (0.01, 0.02, 0.04) when increasing the training set size by around a factor of 4 (from LRS3 to LRS3+Vox2-en).

