# OpenReview forum: "Jointly Learning Visual and Auditory Speech Representations from Raw Data"
_ICLR.cc/2023/Conference — ICLR 2023 poster_

### Official Review · Reviewer_CL4k · 2022-10-25

**Confidence:** 4
**Correctness:** 4
**Technical Novelty And Significance:** 3
**Empirical Novelty And Significance:** 3
**Recommendation:** 8

**Clarity, Quality, Novelty And Reproducibility:**

The paper is clearly written and well structured.  The experiments are thorough, and the results are impressive.  Visual-only speech recognition is challenging, so results with percentages in the low twenties beats (expert) human performance — humans require other context other than just a view of the lips to understand speech (actual recognition rates are incredible low).

**Strength And Weaknesses:**

+The paper is well written.

+Thorough evaluation of the approach.

+Impressive results beating existing state of the art.

-The main text refers the reader to Appendix A.3 for details of the language model, but there is no discussion of what the language model actually is.  There is a term in the overall loss,  but it is not clear how L_lm is calculated.

-Are numbers available about run-time complexity?

-The learned representations are applied only to A/V speech recognition -- it would be useful to show their utility in a wider range of AV tasks (although this is listed as future work),


**Summary Of The Paper:**

The focus of this work is learning representations of auditory and visual speech from the underlying raw signals, as opposed to learning a representation from hand-crafted features as is typically done.  The experiments include an ablation over design choices.  The combination of within-modality modeling (for audio) and cross-modality modeling (for audio and vision) provide a powerful representation, as demonstrated by applying to automatic speech recognition and visual speech recognition.  The results are impressive and set a new state of the art.

**Summary Of The Review:**

This paper proposes new representations of auditory and visual speech that lead to very low WERs in each domain.  The results are impressive and the experiments clearly test the contribution of the different components of the system.

---

> ### Author Response · Authors · 2022-11-13
> **Response to Reviewer CL4k**
>
> Thank you for your comments and time. Below we address the key issues raised.
>
> > The main text refers the reader to Appendix A.3 for details of the language model, but there is no discussion of what the language model actually is. There is a term in the overall loss, but it is not clear how L_lm is calculated.
>
> Thank you for bringing this to our attention. We have now updated Appendix A.4 (formerly A.3) to make the fine-tuning / decoding procedure clearer. Specifically, we use a 16-block Transformer-based language model, as proposed in [1]. The hidden size/MLP size/attention heads are 512/2048/8. During decoding, and when using a language model, we incorporate the weighted prior score S_LM (see subsection “CTC/attention decoding” in Appendix A.4) through shallow fusion, along with the scores from the CTC and attention branches, as described in [2].
>
> > Are numbers available about run-time complexity?
>
> It takes around 15 minutes / 1 hour per epoch to pre-train our Base models on LRS3 / LRS3+Vox2-en, and 2 hours per epoch for our Large model on LRS3+Vox2-en. Fine-tuning Base (for both VSR and ASR) takes around 30 seconds per epoch in the low-resource setting and 7 minutes per epoch in the high-resource setting. Fine-tuning Large takes approximately 1 minute and 20 minutes per epoch in the low- and high-resource setting, respectively. Inference on one A100 GPU (without batching) takes around 3 seconds to decode 10 seconds of footage for Base and 5 seconds for Large. We have now added these details to Appendices A.3 and A.4.
>
> > The learned representations are applied only to A/V speech recognition -- it would be useful to show their utility in a wider range of AV tasks (although this is listed as future work)
>
> Indeed, we look forward to applying RAVEn to multiple tasks beyond speech recognition. As an initial attempt along this direction, we apply RAVEn to the task of video-to-speech synthesis, which aims to output the speech waveform given a silent video of lip movements. We follow the protocol of SVTS [3], the current state-of-the-art method, to perform video-to-speech on the LRS3 test set after training on LRS3+Vox2-en. We study the effect of initialising the video encoder with the weights from our Large pre-trained model.
>
> | Method        | Encoder     | Training dataset | PESQ     | STOI     | ESTOI    |
> |---------------|-------------|------------------|----------|----------|----------|
> | SVTS          | Conformer   | LRS3             | 1.25     | 0.51     | 0.27     |
> | SVTS          | Conformer   | LRS3+Vox2-en     | 1.26     | 0.53     | 0.31     |
> | SVTS          | Transformer | LRS3+Vox2-en     | 1.26     | 0.53     | 0.30     |
> | SVTS w/ RAVEn | Transformer | LRS3+Vox2-en     | **1.30** | **0.56** | **0.36** |
>
> As can be seen in the Table, RAVEn pre-training significantly surpasses training from scratch, as well as SVTS that uses a Conformer encoder. Thus, with very minimal tuning, using RAVEn allows us to set a new state-of-the-art for video-to-speech synthesis on the challenging LRS3 dataset. In response to your comment, we have now added these results, as well as more technical details, in Appendix E.
>
> [1] Irie, Kazuki, Albert Zeyer, Ralf Schlüter, and Hermann Ney. "Language modeling with deep transformers." arXiv preprint arXiv:1905.04226 (2019).
>
> [2] Watanabe, Shinji, Takaaki Hori, Suyoun Kim, John R. Hershey, and Tomoki Hayashi. "Hybrid CTC/attention architecture for end-to-end speech recognition." IEEE Journal of Selected Topics in Signal Processing 11, no. 8 (2017): 1240-1253.
>
> [3] Mira, R., Haliassos, A., Petridis, S., Schuller, B.W. and Pantic, M., 2022. SVTS: Scalable Video-to-Speech Synthesis. arXiv preprint arXiv:2205.02058.

---

> > ### Comment · Reviewer_CL4k · 2022-11-14
> > **Thanks for clarifying**
> >
> > Thanks for providing the additional content/context to address my questions.

---

### Official Review · Reviewer_XgXU · 2022-10-25

**Confidence:** 4
**Clarity, Quality, Novelty And Reproducibility:** The paper is clear, and has a good qu…
**Correctness:** 3
**Technical Novelty And Significance:** 3
**Empirical Novelty And Significance:** 3
**Recommendation:** 6

**Strength And Weaknesses:**

Strengths:

1- The paper tackles an important and timely problem, and the solution is relatively novel.

2- The paper is well-written and structured.

3- It achieves strong results.

4- Implementation details and hyperparameters are well-presented, and the authors aim to make the code public.

Weaknesses:

1- Some editing issues are still present in the paper. For example, the heading "Introduction" is missing after the abstract.

2- The datasets that have been used could have been expanded

3- As the paper mentions, the setup is asymetric in terms of the two modalities, i.e., the auditory stream predicts both the visual and auditory targets, but the visual one predicts only the auditory targets. It is perfectly fine to have a final solution thagt is asymetric. But it would have been nice to include experiments, e.g. cross-modal + video within modal is missing.

4- I understand the novelty, but it would have been nice to include a bit more intuition regarding the results and experiments to show why the performance is obtained. E.g., how does the cross-modal loss result in better representations?

5- From what I understand, the downstream is done on multimodal. How is the performance on unimodal? I.e., does the teacher-student approach result in better unimodal learning too?

* Update: after the rebuttal, I have now increases my score.


**Summary Of The Paper:**

The paper proposes a merthod called RAVEn, which is a self-supervised multi-modal approach to jointly learn visual and auditory speech representations. The paper uses masked inputs and momentum encoders. The proposed approach is asymmetric in terms of the two modalities and their pretext tasks. Strong results are obtained.

**Summary Of The Review:**

Please see my reviews above

---

> ### Author Response · Authors · 2022-11-13
> **Response to Reviewer XgXU (Part 1)**
>
> Thank you for your comments and time. Below we address the key issues raised.
>
> > the heading "Introduction" is missing after the abstract.
>
> Thank you for pointing this out. This has now been fixed.
>
> > The datasets that have been used could have been expanded.
>
> We fine-tune our models on the LRS3 and LRS2 datasets (as well as a subset of LRS3 for our low-resource labelled data setting) to compare with the state-of-the-art approaches, which have been similarly tested. We would like to emphasise that LRS3 and LRS2 are by far the largest, most challenging, and most competitive publicly available transcribed audio-visual datasets for continuous speech recognition, featuring diverse speakers, an unconstrained vocabulary, and long sentences. In terms of pre-training, we use VoxCeleb2, one of the largest and most popular public non-transcribed audio-visual datasets; this allows us to fairly compare with AV-HuBERT, which also uses VoxCeleb2 for pre-training. We have now added more details on the datasets used in Appendix A.1.
>
> > As the paper mentions, the setup is asymetric in terms of the two modalities, i.e., the auditory stream predicts both the visual and auditory targets, but the visual one predicts only the auditory targets. It is perfectly fine to have a final solution thagt is asymetric. But it would have been nice to include experiments, e.g. cross-modal + video within modal is missing.
>
> Thank you for the suggestion. We agree that including the cross-modal + video within-modal experiment in Table 3 is useful. We show a version of the updated table below.
>
> | Setting                     | VSR (WER %) | ASR (WER %) |
> |-----------------------------|-------------|-------------|
> | Within-modal                | 92.7        | 15.5        |
> | Cross-modal                 | 40.8        | 14.0        |
> | Cross- + within-modal       | 49.0        | 14.0        |
> | Cross- + video within-modal | 55.3        | 16.4        |
> | Cross- + audio within-modal | **32.9**    | **12.2**    |
>
> It is evident that cross-modal + video within-modal performs much worse than cross-modal + audio within-modal for both VSR and ASR, indicating that the asymmetry only works one way, which is consistent with our intuitions and analysis. We have also updated Figure 2 to include this setting.
>
> > I understand the novelty, but it would have been nice to include a bit more intuition regarding the results and experiments to show why the performance is obtained. E.g., how does the cross-modal loss result in better representations?
>
> Thank you for this suggestion, which prompted us to provide more intuition. The cross-modal loss forces the video network to encode information predictive of the audio representations, and vice versa. Since lip movements and the corresponding speech waveform are correlated in terms of lexical content (i.e., which words were uttered), the resulting representations are expected to capture rich semantic information related to speech. At the same time, the representations are likely to be, to a large extent, invariant to factors unshared between the two modalities, such as visual or auditory noise. This is more so than in uni-modal learning, where a masked and an unmasked sample overlap greatly in terms of noise. These properties of cross-modal learning are expected to benefit generalisation during the fine-tuning phase, especially in the low-resource environment, and we verify this empirically in Table 3. We have now augmented the relevant discussion in Section 4.4 (“Prediction tasks”) to include these points.

---

> > ### Author Response · Authors · 2022-11-13
> > **Response to Reviewer XgXU (Part 2)**
> >
> > > From what I understand, the downstream is done on multimodal. How is the performance on unimodal? I.e., does the teacher-student approach result in better unimodal learning too?
> >
> > We note that pre-training is multi-modal (in the sense that the visual and auditory encoders predict targets from the other modality), but the resulting encoders are uni-modal, so the downstream (fine-tuning) results are for uni-modal VSR / ASR. In response to the request of Reviewer Hokd, we have now also added results in Appendix C.3 for audio-visual speech recognition by fusing the representations outputted by the two encoders. We find that audio-visual fine-tuning results in very marginal improvements compared to audio-only training in clean (non-noisy) environments.
> >
> > In terms of pre-training, a uni-modal pretext task results in very poor visual representations but decent auditory representations, as shown in Table 3 (evaluated on the val set) and discussed in Section 4.4. We compare here uni-modal pre-training with the audio-only HuBERT method on the LRS3 test set in the low-resource setting, with LRS3 pre-training. We use our Base model for audio-only pre-training and report the HuBERT result using the larger Base model (as reported in AV-HuBERT), which contains roughly double the number of parameters of our Base variant.
> >
> > | Method                     | ASR (WER %) |
> > |----------------------------|-------------|
> > | HuBERT                     | 5.4         |
> > | Audio-only teacher-student | 5.4         |
> > | RAVEn                      | **4.7**     |
> >
> >
> > It is evident that the teacher-student audio-only pre-training yields similar results (but with less parameters) to HuBERT. However, our proposed RAVEn clearly outperforms both.

---

> > > ### Comment · Reviewer_XgXU · 2022-11-29
> > > **Reply**
> > >
> > > I appreciate the authors' efforts to address my comments. I find the responses satisfactory and convincing, and will accordingly increase my score.

---

### Official Review · Reviewer_Hokd · 2022-10-25

**Confidence:** 4
**Correctness:** 4
**Technical Novelty And Significance:** 1
**Empirical Novelty And Significance:** 2
**Recommendation:** 6

**Clarity, Quality, Novelty And Reproducibility:**


Taking good practices in existing SSL approaches, the proposed method achieves competitive performance on ASR and VSR tasks using the learned representations. But, the paper novelty on the technical side is very marginal.

The authors provided implementation details, which can help reproduce the proposed method. They promised the code and pre-trained models would be publically available.

**Strength And Weaknesses:**

Pros:

\+  The proposed cross-modal learning mechanism is well-motivated. Considering that audio contains more information relevant to speech than visual, the authors propose a learning strategy that accounts for the modality difference. Its audio student predicts outputs from both audio and video teacher, whereas the video student predicts only audio teacher outputs.

\+ Extensive experiments and ablation studies in terms of ASR and VSR tasks can validate the effectiveness of the proposed multi-modal self-supervised approach.


Cons:

\- The technical novelty is limited. I did not find any novel and significant technical components in the proposed approach. The key designs in the proposed approach, such as SSL with Knowledge Distillation, masked prediction, and momentum-based teacher, can be found in previous works. To me, the proposed RAVEn is like an audio-visual version of the Self-Supervised Vision Transformer: DINO [1].

[1] Mathilde Caron, Hugo Touvron, Ishan Misra, Herv´e J´egou, Julien Mairal, Piotr Bojanowski, and Armand Joulin. Emerging properties in self-supervised vision transformers. In Proceedings of the 18th IEEE/CVF International Conference on Computer Vision (ICCV), pp. 9650–9660, 2021.

\-  In real-world videos, audio and visual content are not always matched. For example, a speaking person can be out of the screen. How can the proposed method defend against the issue?

\- Since the proposed method can jointly learn audio and visual representations, it is straightforward to perform audio-visual speech recognition using the learned representations. But, in the paper, only ASR and VSR are explored. In addition, we only see very small improvements in the ASR task (LRS3 high-resource setting).

**Summary Of The Paper:**

In this paper, the authors propose a self-supervised multi-modal approach: RAVEn to jointly learn visual and speech representations. The key components in RAVEn take good practices (e.g., masked prediction, Transformer, and Knowledge Distillation) in existing self-supervised learning approaches. But differently, it leverages audio-visual cross-modal correspondence to train the teacher-student learning model. Extensive experiments show that RAVEn can help to learn powerful visual and auditory speech representations for visual and auditory speech recognition.

**Summary Of The Review:**

Most of the key components in the proposed method, such as SSL with Knowledge Distillation, masked prediction, momentum-based teacher, and audio-visual SSL, can be found in previous approaches. Due to the limited technical novelty, my current rating for this paper is reject.

***Post-Rebuttal***

Thank the authors for responding to my questions! Most of my concerns have been clarified by the rebuttal. Although the technical contributions are incremental, the proposed RAVEn can be a strong baseline for joint visual-speech representation learning.

My rating is upgraded to 6 from 3.

---

> ### Author Response · Authors · 2022-11-13
> **Response to Reviewer Hokd (Part 1)**
>
> Thank you for your comments and time. Below we address the key issues raised.
>
> > The technical novelty is limited. I did not find any novel and significant technical components in the proposed approach. The key designs in the proposed approach, such as SSL with Knowledge Distillation, masked prediction, and momentum-based teacher, can be found in previous works. To me, the proposed RAVEn is like an audio-visual version of the Self-Supervised Vision Transformer: DINO.
>
> Isolated aspects of RAVEn have indeed been inspired by previous self-supervised work in other domains, as described in our Related Works section. Nevertheless, we believe that there is substantial technical novelty in RAVEn’s design taken as a whole, and it is what enabled us to achieve state-of-the-art results for self-supervised audio-visual speech representation learning using only raw data and a single pre-training stage / phase.
>
> Specifically, there is similarity between RAVEn and DINO in the use of momentum encoders to generate targets (we note that momentum encoders were used for self-supervised image representation learning even before DINO, e.g., in MoCo [2] and BYOL [3]). However, in addition to being multi-modal and having been developed for a substantially different domain / task, RAVEn exhibits significant methodological differences from DINO. In essence, DINO defines its pretext task by feeding the student and teacher networks different views of the same image via domain-specific augmentations such as random cropping and colour jittering. In contrast, RAVEn does not rely on domain-specific augmentations but rather on a combination of cross-modal learning and masked prediction to drive representation learning for visual and auditory speech signals. Predicting auditory targets from video and vice versa forces the encoders to capture the highly semantic information shared between the two (synchronised) modalities for talking faces, such as lexical content, which would be difficult to achieve through hand-crafted augmentations. Masked prediction enforces context understanding, a crucial aspect of effective speech recognition.
>
> There are also numerous additional technical differences, which are vital to the performance, such as:
> * DINO outputs global representations (through a [cls] token), i.e., one embedding per image. RAVEn outputs and predicts frame-wise representations, due to the time-varying nature of speech / lip movements and the fine-grained temporal synchronicity between the signals.
> * Unlike DINO, RAVEn uses lightweight Transformer predictors / heads for the students, allowing separate heads for cross- and within-modal prediction. Further, it enables the application of mask tokens only on the predictors (and not the encoders) to reduce the input discrepancy between pre-training and fine-tuning (Section 4.4).
> * DINO uses a cross-entropy loss with centering and sharpening operations to prevent representation collapse, whereas RAVEn uses a cosine distance loss.
> * RAVEn uses a specific asymmetric combination of within- and cross-modal losses, shown to be suited for visual/auditory speech representation learning (Table 3).
>
> We have now made the main differences between RAVEn and DINO clearer in Section 2.
>
> > In real-world videos, audio and visual content are not always matched. For example, a speaking person can be out of the screen. How can the proposed method defend against the issue?
>
> RAVEn exploits the correspondence and synchronicity between speech and lip movements. As other related audio-visual methods, it assumes that (1) the videos contain a speaking face and that (2) there is audio-visual alignment. If these conditions are not met for specific samples in a dataset, automated tools like face/voice activity detectors can be used to remove those samples, and off-the-shelf audio-visual synchronisation models can ensure that the video and audio are aligned.
>
> [2] He, Kaiming, Haoqi Fan, Yuxin Wu, Saining Xie, and Ross Girshick. "Momentum contrast for unsupervised visual representation learning." In Proceedings of the IEEE/CVF conference on computer vision and pattern recognition, pp. 9729-9738. 2020.
>
> [3] Jean-Bastien Grill, Florian Strub, Florent Altche, Corentin Tallec, Pierre Richemond, Elena ´ Buchatskaya, Carl Doersch, Bernardo Avila Pires, Zhaohan Guo, Mohammad Gheshlaghi Azar, et al. Bootstrap your own latent-a new approach to self-supervised learning. In Proceedings of the 33rd Advances in Neural Information Processing System (NIPS), volume 33, pp. 21271–21284, 2020.

---

> > ### Author Response · Authors · 2022-11-13
> > **Response to Reviewer Hokd (Part 2)**
> >
> > > Since the proposed method can jointly learn audio and visual representations, it is straightforward to perform audio-visual speech recognition using the learned representations. But, in the paper, only ASR and VSR are explored.
> >
> > Thank you for this point. We agree that it is possible to use the learned auditory and visual representations for audio-visual speech recognition. To that end, following [4], we concatenate the outputs of the two encoders, feed the resulting embeddings to a 2-layer MLP module, and then fine-tune for speech recognition. We have included the results and more technical details in Appendix C.3. Below are the results in the low- and high-resource settings for the Large model with LRS3+Vox2-en pre-training (without self-training).
> > | Fine-tune setting | Resource | WER (%) |
> > |-------------------|----------|---------|
> > | Audio-only        | Low      | 2.6     |
> > | Audio-visual      | Low      | 2.5     |
> > | Audio-only        | High     | 1.4     |
> > | Audio-visual      | High     | 1.4     |
> >
> > As can be seen, audio-visual training gives marginal improvements (if any) compared to audio-only when the audio is clean (i.e., not noisy), consistent with findings in [4]. Investigating the impact of audio-visual training and testing on audio corrupted with various noise types, is outside the scope of this work, but it will be interesting to take up in future work.
> >
> > > we only see very small improvements in the ASR task (LRS3 high-resource setting)
> >
> > Indeed, we are on par with AV-HuBERT in the ASR LRS3 high-resource setting (which is arguably the setting least sensitive to pre-training, as seen by comparing training from scratch to pre-training in Tables 1 and 2). However, we simultaneously achieve state-of-the-art results in the self-supervised VSR LRS3 high-resource setting (23.4% vs 26.9% WER) using the same pre-training phase (in contrast to AV-HuBERT, which uses separate pre-training phases for VSR and ASR). We further point out that our work additionally sets new state-of-the-art results in the self-supervised VSR (24.4% vs 28.6% WER) and ASR (1.9% vs 2.9% WER) LRS3 low-resource settings, as well as on LRS2 (Appendix B), both for VSR (18.6% vs 22.6%) and ASR (2.1% vs 2.7%).
> >
> > [4] Pingchuan Ma, Stavros Petridis, and Maja Pantic. End-to-end audio-visual speech recognition with conformers. In Proceedings of the 46th IEEE International Conference on Acoustics, Speech and Signal Processing (ICASSP), pp. 7613–7617, 2021b

---

> > > ### Comment · Reviewer_Hokd · 2022-11-13
> > > **Follow-up questions**
> > >
> > > Thank the authors for responding to my concerns!
> > >
> > > Regarding the new results, can you give more analysis to explain why audio-visual joint inference cannot help speech recognition? From the answer to Q2, we can learn that the method only learns from clean audio and visual pairs. Whether is this a possible reason?

---

> > > > ### Author Response · Authors · 2022-11-14
> > > > **Response to follow-up**
> > > >
> > > > Thank you for the follow-up.
> > > >
> > > > The addition of the visual modality often leads to negligible performance improvements in clean (non-noisy) conditions, as also observed by previous works on audio-visual speech recognition [4-6]. For example, [4] (reference in previous reply) report 2.3% WER for both their audio-only and audio-visual models on LRS3, and [5] report 1.6% for both. We believe this is due to two main reasons: 1) it is much easier to perform speech recognition from a speech waveform than from lip movements (and the lip movements do not provide much extra information) in clean conditions, and 2) the word error rate is already very low, which makes it difficult to achieve noticeable improvements.
> > > >
> > > > On the other hand, when the audio input is noisy, then the visual modality does indeed noticeably help. We demonstrate this here by adding babble noise of varying signal-to-noise ratio (SNR) from the NOISEX corpus [7] to the LRS3 test set, and comparing on the noisy data our audio-only model with the audio-visual one (both of which were pre-trained and fine-tuned using clean audio).
> > > >
> > > > |              | Clean   | SNR=5dB   | SNR=0dB   | SNR=-5dB   |
> > > > |--------------|---------|---------|---------|----------|
> > > > | Audio-only   | **1.4** | 5.1     | 13.7    | 68.9     |
> > > > | Audio-visual | **1.4** | **4.6** | **6.7** | **24.8** |
> > > >
> > > > It is evident that as the audio noise level increases (and the signal-to-noise ratio reduces), the visual modality becomes increasingly important and the gap between audio-only and audio-visual performance widens.
> > > >
> > > > [5] Serdyuk, Dmitriy, Otavio Braga, and Olivier Siohan. "Transformer-Based Video Front-Ends for Audio-Visual Speech Recognition." arXiv preprint arXiv:2201.10439 (2022).
> > > >
> > > > [6] Makino, Takaki, Hank Liao, Yannis Assael, Brendan Shillingford, Basilio Garcia, Otavio Braga, and Olivier Siohan. "Recurrent neural network transducer for audio-visual speech recognition." In 2019 IEEE automatic speech recognition and understanding workshop (ASRU), pp. 905-912. IEEE, 2019.
> > > >
> > > > [7] Varga, Andrew, and Herman JM Steeneken. "Assessment for automatic speech recognition: II. NOISEX-92: A database and an experiment to study the effect of additive noise on speech recognition systems." Speech communication 12, no. 3 (1993): 247-251.

---

> > > > > ### Comment · Reviewer_Hokd · 2022-11-15
> > > > > **Response**
> > > > >
> > > > > Nice explanation. Most of my concerns have been clarified. I am happy to upgrade my rating.

---

> > > > > > ### Comment · Program_Chairs · 2022-11-28
> > > > > > **msg from SPC**
> > > > > >
> > > > > > authors - can you also comment on ethics concern raised?
> > > > > >
> > > > > > AC - if the issue isn't resolved, please bring it back to us.

---

> > > > > > > ### Author Response · Authors · 2022-11-30
> > > > > > > **Response to Ethics Concern**
> > > > > > >
> > > > > > > > authors - can you also comment on ethics concern raised?
> > > > > > >
> > > > > > > As no ethics concerns were raised by Reviewer Hokd (which this discussion thread corresponds to), we assume that this refers to Reviewer ykA5’s concern. We have added a comment addressing this in Reviewer ykA5’s thread, which we reproduce below along with the original quoted concern.
> > > > > > >
> > > > > > > > I am unsure of this, but I believe that this work used YouTube videos as training data and thus requires downloading them which is against YouTube's term of service. There has been a lot of published work though that has used YouTube as a data source such as AudioSet [1] and VoxCeleb [2]. [1] is even from Google, YouTube's parent.
> > > > > > >
> > > > > > > We would like to point out that we did not download any videos from YouTube for this work and only used datasets that were made public for research purposes, i.e., VoxCeleb2 [1], LRS3 [2], and LRS2 [3]. In particular, we used the cropped face .mp4 videos that were available on the official dataset webpages and which complied with the following licenses: Creative Commons BY-NC-ND 4.0 license and Creative Commons Attribution 4.0 International License, TED terms of use, and BBC’s terms of use. We also note that these datasets are widely used in the community (e.g., VoxCeleb2 has around 1.5k citations), including by recent works such as [4-6].
> > > > > > >
> > > > > > > We will make sure to include these points in the ethics statement of our paper when we are able to upload a revised version.
> > > > > > >
> > > > > > > [1] Joon Son Chung, Arsha Nagrani, and Andrew Zisserman. “Voxceleb2: Deep speaker recognition.” In
> > > > > > > Proceedings of the 19th Annual Conference of International Speech Communication Association (INTERSPEECH), 2018.
> > > > > > >
> > > > > > > [2] Afouras, Triantafyllos, Joon Son Chung, and Andrew Zisserman. "LRS3-TED: a large-scale dataset for visual speech recognition." arXiv preprint arXiv:1809.00496 (2018).
> > > > > > >
> > > > > > > [3] Son Chung, Joon, Andrew Senior, Oriol Vinyals, and Andrew Zisserman. "Lip reading sentences in the wild." In Proceedings of the IEEE conference on computer vision and pattern recognition (CVPR),  2017.
> > > > > > >
> > > > > > > [4] Bowen Shi, Wei-Ning Hsu, Kushal Lakhotia, and Abdelrahman Mohamed. “Learning audio-visual speech representation by masked multimodal cluster prediction.” In Proceedings of the 10th International Conference on Learning Representations (ICLR), 2022.
> > > > > > >
> > > > > > > [5] K. R. Prajwal, Triantafyllos Afouras, and Andrew Zisserman. “Sub-word level lip reading with visual attention.” In Proceedings of the 35th IEEE/CVF Conference on Computer Vision and Pattern Recognition (CVPR), 2022.
> > > > > > >
> > > > > > > [6] Mira, Rodrigo, Alexandros Haliassos, Stavros Petridis, Björn W. Schuller, and Maja Pantic. "SVTS: Scalable Video-to-Speech Synthesis." In Proceedings of the 23rd Annual Conference of International Speech Communication Association (INTERSPEECH), 2022.

---

### Official Review · Reviewer_ykA5 · 2022-10-26

**Confidence:** 4
**Correctness:** 3
**Technical Novelty And Significance:** 3
**Empirical Novelty And Significance:** 3
**Recommendation:** 6

**Clarity, Quality, Novelty And Reproducibility:**

The paper is clearly written and of good quality. The novelty of the task is low since similar work has been done, but the chosen method is novel in that previous work in this area required multi stages of clustering with hand crafted features before learning representations that can use the raw audio and video features. This work introduces an elegant one stage approach, and provides many experimental details that would aid in the reproducibility in the work. Overall, the technique and provided details should yield results that others can replicate, but not having published scripts and code yet, hurts reproducibility at the moment (the author's promise to release these upon publication).

One claim of the paper is not correct. "surpassing... recent fully-supervised method trained on 90,000 hours of non-public labelled data." These works from Deepmind/Google Shillingford et al. (2019), Makino et al. (2019), Serdyuk et al. (2021) and Serdyuk et al. (2022) all use found data for *semi-supervised* training ie. a supervised trained recognizer, trained on limited data, is used to produce labels from user uploaded captions as described in "Large scale deep neural network acoustic modeling with semi-supervised training data for YouTube video transcription", H. Liao et al, ASRU 2013. These is not hand labeled data in the supervised sense---user uploaded captions can be errorful and so the supervised model is used to validate them for use as labels.



**Details Of Ethics Concerns:**

I am unsure of this, but I believe that this work used YouTube videos as training data and thus requires downloading them which is against YouTube's term of service. There has been a lot of published work though that has used YouTube as a data source such as AudioSet [1] and VoxCeleb [2]. [1] is even from Google, YouTube's parent.

**Strength And Weaknesses:**

Strengths:
- the paper presents a different approach to previous work on learning audio-visual representations in an unsupervised manner compared to the work in Shi ICLR 2022. In Shi's AV-HUBERT, they required a complicated multistage process of first using MFCC features to cluster, before learning an AV representation from raw data. In this work, a one-stage training regime is achieved.
- the provides extensive analyses on the various aspects of their approach. of note, the ablations in Table 3 nicely motivate the use of asymmetric student-teacher training for best results.
- the paper achieves state-of-the-art results for unsupervised training for visual-speech-recognition on TED-LRS3

Weaknesses
- from a certain point of view, the paper is incremental over Shi's ICLR 2022 work. it achieves similar results, however in a simpler manner.
- the difference of 1.9% in this paper for TED-LRS is a significantly off from the 1.3% found in Shi 2022.

**Summary Of The Paper:**

The paper explores a method for pretraining an Audio-Visual speech recognition model directly from raw video with both audio and visual signals. It does so by asymmetrically applying two student-teacher networks 1. the audio student learns to predict both audio and visual targets generated by respective teachers, 2. the video student only learns to predict the audio features. The rationale for this is that the audio provides a much stronger signal than the video. The results are competitive with state-of-the-art for self-supervised learning with small amount of fine-tuning data, e.g. for VASR on TED-LRS3 24.4% better vs 26.9% (Shi ICLR 2022), and for AVASR on TED-LRS3 1.9% is worse than the 1.3% in Shi ICLR 2022.

**Summary Of The Review:**

Overall the paper makes a nice contribution to the growing area of unsupervised training on Audio-Visual data. They provide an incremental way of training an AV model in a simpler model than in the past and achieve results comparable to the state-of-the-art so I support accepting this paper for publication at ICLR.

---

> ### Author Response · Authors · 2022-11-13
> **Response to Reviewer ykA5**
>
> Thank you for your comments and time. Below we address the key issues raised.
>
> > for VASR on TED-LRS3 24.4% better vs 26.9% (Shi ICLR 2022), and for AVASR on TED-LRS3 1.9% is worse than the 1.3% in Shi ICLR 2022.
>
> We believe this is a misunderstanding regarding the comparisons between the results for RAVEn and Shi ICLR 2022 (AV-HuBERT). These quoted results for RAVEn, 24.4% WER for VSR / 1.9% for ASR, are from our low-resource labelled data setting in Table 1 (fine-tuning on only 30 hours of LRS3), whereas those quoted for AV-HuBERT, 26.9% VSR / 1.3% ASR, are from the high-resource setting in Table 2 (fine-tuning on the full 433-hour LRS3 dataset). In fact, as shown in Table 1, our best result for ASR in low-resource, 1.9% WER, is significantly better than AV-HuBERT’s best in the same setting, 2.9% WER; a similarly significant improvement is observed for VSR in low-resource: RAVEn’s 24.4% WER vs AV-HuBERT’s 28.6% WER. In terms of the high-resource setting (Table 2), we again outperform AV-HuBERT for VSR (23.4% vs 26.9%) and are on par with AV-HuBERT for ASR (1.4% vs 1.3%).
>
> > from a certain point of view, the paper is incremental over Shi's ICLR 2022 work. it achieves similar results, however in a simpler manner.
>
> We agree that our approach is simpler than Shi's ICLR 2022, but, having clarified the misunderstanding above, we also believe that RAVEn pushes the state of the art for audio-visual speech representation learning. Our best results outperform AV-HuBERT in the low-resource VSR and ASR settings by 15% and 34% in relative terms, respectively, and in the high-resource VSR setting by 13%. We agree that RAVEn achieves similar results to AV-HuBERT in the high-resource ASR setting, but this is arguably the setting least sensitive to pre-training, as evidenced by comparing training from scratch to pre-training in Tables 1 and 2.
>
> > One claim of the paper is not correct. "surpassing... recent fully-supervised method trained on 90,000 hours of non-public labelled data." These works from Deepmind/Google Shillingford et al. (2019), Makino et al. (2019), Serdyuk et al. (2021) and Serdyuk et al. (2022) all use found data for semi-supervised training ie. a supervised trained recognizer, trained on limited data, is used to produce labels from user uploaded captions as described in "Large scale deep neural network acoustic modeling with semi-supervised training data for YouTube video transcription", H. Liao et al, ASRU 2013. These is not hand labeled data in the supervised sense---user uploaded captions can be errorful and so the supervised model is used to validate them for use as labels.
>
> Thank you for pointing this out. We have now corrected this in the paper by removing any mentions of “fully-supervised” when referring to that work, and we have clarified in the caption of Table 1 that some of the non-publicly available data that methods use is not hand-labelled.

---

> > ### Author Response · Authors · 2022-11-30
> > **Response to Ethics Concern**
> >
> > As prompted by the Program Chairs, we address the following concern:
> > > I am unsure of this, but I believe that this work used YouTube videos as training data and thus requires downloading them which is against YouTube's term of service. There has been a lot of published work though that has used YouTube as a data source such as AudioSet [1] and VoxCeleb [2]. [1] is even from Google, YouTube's parent.
> >
> > We would like to point out that we did not download any videos from YouTube for this work and only used datasets that were made public for research purposes, i.e., VoxCeleb2 [1], LRS3 [2], and LRS2 [3]. In particular, we used the cropped face .mp4 videos that were available on the official dataset webpages and which complied with the following licenses: Creative Commons BY-NC-ND 4.0 license and Creative Commons Attribution 4.0 International License, TED terms of use, and BBC’s terms of use. We also note that these datasets are widely used in the community (e.g., VoxCeleb2 has around 1.5k citations), including by recent works such as [4-6].
> >
> > We will make sure to include these points in the ethics statement of our paper when we are able to upload a revised version.
> >
> > [1] Joon Son Chung, Arsha Nagrani, and Andrew Zisserman. “Voxceleb2: Deep speaker recognition.” In
> > Proceedings of the 19th Annual Conference of International Speech Communication Association (INTERSPEECH), 2018.
> >
> > [2] Afouras, Triantafyllos, Joon Son Chung, and Andrew Zisserman. "LRS3-TED: a large-scale dataset for visual speech recognition." arXiv preprint arXiv:1809.00496 (2018).
> >
> > [3] Son Chung, Joon, Andrew Senior, Oriol Vinyals, and Andrew Zisserman. "Lip reading sentences in the wild." In Proceedings of the IEEE conference on computer vision and pattern recognition (CVPR),  2017.
> >
> > [4] Bowen Shi, Wei-Ning Hsu, Kushal Lakhotia, and Abdelrahman Mohamed. “Learning audio-visual speech representation by masked multimodal cluster prediction.” In Proceedings of the 10th International Conference on Learning Representations (ICLR), 2022.
> >
> > [5] K. R. Prajwal, Triantafyllos Afouras, and Andrew Zisserman. “Sub-word level lip reading with visual attention.” In Proceedings of the 35th IEEE/CVF Conference on Computer Vision and Pattern Recognition (CVPR), 2022.
> >
> > [6] Mira, Rodrigo, Alexandros Haliassos, Stavros Petridis, Björn W. Schuller, and Maja Pantic. "SVTS: Scalable Video-to-Speech Synthesis." In Proceedings of the 23rd Annual Conference of International Speech Communication Association (INTERSPEECH), 2022.

---

> ### Author Response · Authors · 2022-12-07
> **Any remaining concerns?**
>
> Thank you again for your helpful comments. As the discussion period is soon coming to an end, we would like to enquire whether there are any remaining concerns. If so, we would be happy to address them.

---

### Author Response · Authors · 2022-11-13
**Summary of Paper Changes**

We thank the reviewers for their thoughtful comments and feedback, which have helped us improve the paper. We are glad that the reviewers appreciated our method’s novelty (Reviewers ykA5 and XgXU) and strong performance (Reviewers XgXU and CL4k), our extensive experiments (Reviewers ykA5, Hokd, CL4k), and the quality of the writing (Reviewers ykA5, XgXU, CL4k).

Due to limited overlap in the reviewers’ concerns, we address the issues raised directly under each review. Given the feedback, we have made the following changes to the paper (coloured in blue in the revised submission):

* We have made modifications to the abstract to better highlight our method’s state-of-the-art performance in different settings.
* We have added audio-visual fine-tuning experiments in Appendix C.3.
* We have added results in Appendix E for using RAVEn pre-training for another task, video-to-speech synthesis, where we achieve state-of-the-art performance. This validates the suitability of the learned representations beyond speech recognition.
* We have more clearly highlighted the key high-level differences between RAVEn and DINO in Section 2.
* We have added more details on the datasets used in Appendix A.1.
* We have included a cross-modal + video within-modal prediction task setting in Figure 2 and Table 3.
* We have provided more intuition on why cross-modal learning is beneficial in Section 4.4 (“Prediction tasks”).
* We have updated Appendix A.4 (formerly A.3) to make the fine-tuning / decoding procedure clearer.
* We have added numbers on training / fine-tuning / inference run-time in Appendices A.3 and A.4.
* We have removed mentions of “fully-supervised” when referring to the work of Serdyuk et al. (2021), and clarified in the caption of Table 1 that some of the non-publicly available data that methods use is not hand-labelled.
* We have fixed a couple editing issues.

We thank the reviewers again and look forward to any further suggestions or discussion.

---

### Decision · Program_Chairs · 2023-01-20

**Decision:**

Accept: poster

**Justification For Why Not Higher Score:**

An interesting, but not ground breaking method. The novelty lies in the design as a whole.

**Justification For Why Not Lower Score:**

Very strong results. It pushes the state of the art for audio-visual speech representation learning.

**Metareview: Summary, Strengths And Weaknesses:**

The paper presents a method to jointly learn visual and auditory representations from the underlying raw audio-visual input signals, in contrast to commonly used methods that learn representations from engineered features. It further uses asymmetric training targets across the two modalities and is a single stage self-supervised method to make the learning process simpler. The paper is well written. Very strong experimental results have been obtained. The novelty of the paper lies in the design as a whole. I agree with the reviewers to accept the paper.


**Note From Pc:**

if the above contains the word "oral" or "spotlight" please see: "oral" presentation means -> notable-top-5% and "spotlight" means -> notable-top-25%. As stated in our emails, we are disassociating presentation type from AC recommendations